# Artificial Intelligence for Personalised Ophthalmology Residency Training

**DOI:** 10.3390/jcm12051825

**Published:** 2023-02-24

**Authors:** George Adrian Muntean, Adrian Groza, Anca Marginean, Radu Razvan Slavescu, Mihnea Gabriel Steiu, Valentin Muntean, Simona Delia Nicoara

**Affiliations:** 1Department of Ophthalmology, “Iuliu Hatieganu” University of Medicine and Pharmacy Emergency County Hospital, 400347 Cluj-Napoca, Romania; 2Department of Computer Science, Technical University of Cluj-Napoca, 400114 Cluj-Napoca, Romania; 3Department of Surgery, MedLife Humanitas Hospital, 400664 Cluj-Napoca, Romania

**Keywords:** diagnosis of retinal conditions, precision education machine learning, contrastive learning

## Abstract

Residency training in medicine lays the foundation for future medical doctors. In real-world settings, training centers face challenges in trying to create balanced residency programs, with cases encountered by residents not always being fairly distributed among them. In recent years, there has been a tremendous advancement in developing artificial intelligence (AI)-based algorithms with human expert guidance for medical imaging segmentation, classification, and prediction. In this paper, we turned our attention from training machines to letting them train us and developed an AI framework for personalised case-based ophthalmology residency training. The framework is built on two components: (1) a deep learning (DL) model and (2) an expert-system-powered case allocation algorithm. The DL model is trained on publicly available datasets by means of contrastive learning and can classify retinal diseases from color fundus photographs (CFPs). Patients visiting the retina clinic will have a CFP performed and afterward, the image will be interpreted by the DL model, which will give a presumptive diagnosis. This diagnosis is then passed to a case allocation algorithm which selects the resident who would most benefit from the specific case, based on their case history and performance. At the end of each case, the attending expert physician assesses the resident’s performance based on standardised examination files, and the results are immediately updated in their portfolio. Our approach provides a structure for future precision medical education in ophthalmology.

## 1. Introduction

As with numerous other medical and surgical specialties, ophthalmology residency training is also reliant on an apprenticeship education model, which was introduced by Halsted in the late 19th century. The competencies during this apprenticeship, which are passed on from the teaching physician to the trainee, include theoretical knowledge, clinical skills, and surgical training.

With the emergence of technological advances in ophthalmology, there is an increase in the number of investigations a patient undergoes and thus in the time per patient consultation. This technological improvement grows parallel with a rise in the medical gained knowledge, with the doubling time of medical knowledge having been predicted to skyrocket from 50 years in 1950 to an astonishing 73 days in 2020 [1]. These advances, both in terms of technology and knowledge, further extend the physician’s and trainee’s workload. While these advancements favor patient care, the educational time between the trainee and the attending physician reviewing the investigations and discussing the cases is limited, threatening the apprenticeship education model.

Artificial intelligence (AI) has started to play an important role in medicine, from simpler tasks, such as screening and diagnosing various conditions, to more complex assignments, such as predicting disease evolution and structure-function correlation. Taking into consideration the rapid advancements mentioned before, we can observe in this grand puzzle of the apprenticeship model a place for AI-guided education.

During ophthalmology residency training, it is important for residents to be exposed to as many eye conditions as possible. Since medical knowledge advances quickly with technology by its side, numerous subspecialties have emerged in ophthalmology, with physicians specializing in subdomains such as cornea and external disease, refractive surgery, glaucoma, medical retina, vitreoretinal surgery, pediatric ophthalmology and strabismus, orbital surgery and oculoplastics, neuro-ophthalmology, uveitis, and low vision rehabilitation. In the near future, we might observe a further subdivision of these subdomains. In many instances, the time residents spend in various subspecialties is unfortunately short in comparison to the expected learning outcomes, and in some training centers, residents do not have access to all of these subspecialties. Medical retina rotation is considered to be one of the toughest, due to the challenging examination of the eye fundus and the various methods of the presentation of retinal conditions (RCs).

In this paper, we propose a solution for enhancing medical retina rotation. In order for residents to make the most of their time in this rotation, patients presenting to the retina clinic should be referred to them based on the residents’ educational needs. Thus, we view this as a matchmaking problem, where we need to match a known set of residents having different training experiences with an unknown set of patients, which is gradually constructed during the rotation.

In order to tackle this matchmaking problem, we developed an AI-based framework that personalizes the cases residents encounter. The framework is built on two components, (1) a deep learning (DL) model and (2) an expert-system-powered case allocation algorithm. Patients visiting the retina clinic will first have a color fundus photograph (CFP) performed and the fundus image will then be computed by the DL model, which will choose a presumptive diagnosis. The diagnosis is then passed to the case allocation algorithm, which uses an expert system to select the resident who would most benefit from the case, based on their case history and performance.

This approach helps standardize the retina rotation and leads to a more tailored residency experience, with all residents achieving the expected educational competencies.

## 2. Classifying Color Fundus Photographs with Deep Learning

### 2.1. Building the Resident  Dataset

To develop the learning model, we collected, combined, and curated CFPs from three publicly available online datasets: (1) Ocular Disease Intelligent Recognition (ODIR) [2,3], (2) Retinal Fundus Multi-Disease Image Dataset (RFMID) [4], and  (3) the public part (1000 CFPs) of the Joint Shantou International Eye Centre (JSIEC) dataset [5].

Ocular Disease Intelligent Recognition is a structured database containing CFPs from 5000 patients with diagnostic keywords annotated by trained readers with quality control management. Shanggong Medical Technology Co., Ltd. collected the CFPs from “real-life” patients visiting different hospitals and medical centers in China. The CFPs were captured using different fundus cameras available in the market, such as Canon, Zeiss, and Kowa. Since the images are for both the left and the right eye, there are annotations on a patient level and also on an individual eye level. On a patient level, each subject (both eyes) was classified into one or more of the following eight labels: normal (N), diabetes (D), glaucoma (G), cataract (C), age-related macular degeneration (A), hypertension (H), pathological myopia (M), and other diseases/abnormalities (O). On an eye level, there are keywords that refer to the diagnosis or retinal finding of each eye separately. We are using these keywords in building our Resident  dataset.

Retinal Fundus Multi-Disease Image Dataset is a public dataset containing 3200 CFPs with annotations from two independent ophthalmologists which were validated by a team leader. Images were captured with three different fundus cameras: TOPCON 3D OCT-2000, TOPCON TRC-NW300, and Kowa VX-10α. The images are classified into one or more of the 45 different retinal conditions found in Appendix B, Table A2.

The publicly available 1.000 CFPs from the Joint Shantou International Eye Centre were collected from the institution’s Picture Archiving and Communication Systems (PACS). The CFPs were captured with a ZEISS FF450 Plus IR Fundus Camera between 2009–2013 and a Topcon TRC-50DX Mydriatic Retinal Camera between 2013 and 2018. The fundus images were manually labelled by 10 teams, each having 2 licensed ophthalmologists, a senior with more than 7 years of experience and an unspecialised ophthalmologist with 3 years of training. When there was a disagreement inside a team or the image was reported as non-classifiable, the image was sent to a panel of five retinal experts for a final decision. When the 10 teams labelled the images in different retinal conditions, they also added a referable label (observation, routine, semi-urgent, or urgent). The CFPs were classified into 39 categories.

The combined Resident dataset for resident training was built by ophthalmologists with medical retina experience who mapped the categories and keywords from each of the three public datasets from Table 1 as follows:

First, we perform an initial mapping, resulting in 10,592 CFPs images classified into 39 classes. The mapping rules appear in Table 2. The first column shows the resulting classes. The second column includes the ODIR keywords used for mapping, while the last two columns mention the original classes from RFMID and JSIEC. In some cases, the mapping is straightforward, e.g., the branch retinal vein occlusion is present in all three datasets and it corresponds to the new class C11, while in other cases, there are small differences in the specificity of the class; e.g., in the new class C1, the Maculopathy class from JSIEC is combined with different forms of age-related macular degeneration from ODIR. Some of the retinal conditions present in the three original datasets were not included in the Resident dataset, and some of the images from the selected categories/keywords were removed due to bad quality. Furthermore, in the Resident dataset, only the images that belong to at least one of the selected classes were included.

Second, to have sufficient images for testing and to avoid an extremely imbalanced dataset, we only keep the classes containing more than 50 images. The filtered dataset contains 9693 images: 7754 in the train set and 1939 in the test set (Table 3). There are 22 categories containing the main retinal conditions, backed by sufficient data for training the model (50 or more images): normal (C0), age-related macular degeneration (C1), diabetic retinopathy (DR) (C2—mild nonproliferative, C3—moderate nonproliferative, C4—severe nonproliferative, C5—proliferative), glaucoma (C6), hypertensive retinopathy (C7), pathological myopia (C8), tessellated fundus (C9), vitreous degeneration (C10), branch retinal vein occlusion (C11), large optic cup (C13), drusen (C14), epiretinal membrane (C15), optic disc edema (C18), myelinated nerve fibers (C19), rhegmatogenous retinal detachment (C22), refractive media opacity (C25), central serous chorioretinopathy (C27), laser spots (C29), and central retinal vein occlusion (C32). Due to the fact that the RFMID dataset includes only a single class for all the stages of diabetic retinopathy, C2, C3, C4, and C5 have been grouped together in the single class DR.

One observation is that the Resident dataset is not limited to images presenting only one retinal condition. For instance, both glaucoma and wet age-related macular degeneration may simultaneously occur in the same image. Technically, the problem is framed as a multilabel task in which for each image, the model provides a probability for each of the 19 classes without the constraint of having their sum equal to 1. In order to show the conditions which co-occur in ODIR examples, Table A1 presents for each new class the ODIR keywords which characterize the images included in that class. For each keyword, the occurrence number is included. Along with the keywords specific to each class, there are keywords that identify other classes. For example, in the ODIR images from C1, there are 58 occurrences of *dry age-related macular degeneration*, 30 of *wet age-related macular degeneration*, but also 5 of *glaucoma*, meaning that in 5 CFPs there is both AMD and glaucoma.

### 2.2. Applying Contrastive Learning on the Resident  Dataset

The classification model is based on supervised contrastive learning [6] (Figure 1). One reason in favor of using contrastive learning is the fact that it enables learning not only from labelled data but also from unlabelled data. Another reason is the fact that contrastive learning has, as a result, an embedding space where images are clustered based on not only their intra-class similarity but also their inter-class similarity, meaning that conditions that have similar aspects are closer—e.g., drusen (C14) is closer to AMD (C1) than to glaucoma (C6).

The encoder is a ReSNeXt50_32x4d [7]. The projection network consists of one dense layer with a ReLU activation function and another linear layer of size 128. The contrastive loss is adapted to the multilabel setting: when the selected anchor is an image with more than one class $\left\{C_{i}\right\}$, the positive examples include all the samples from the batch which belong to any of the classes from the set $\left\{C_{i}\right\}$. For example, if the anchor belongs to both C1 and C6, then any image of type C1 or C6 is considered a positive sample for this anchor, including the ones which are C1 but not C6, or C6 but not C1.

The input images of size 224×224 were preprocessed by following the steps of Cen et al. [5]. First, the borders of the fundus images were removed, then the eye area was identified with HoughCircles, and finally the image was cropped to that area. Additionally to the preprocessing steps of Cen et al., we applied the Contrast Limited Adaptive Histogram Equalisation (CLAHE) transform. With CLAHE, we aimed to prevent the limitation of the AHE method to over amplify noise in relatively homogeneous regions.

For enhancing training, basic random augmentations are used: scale with range [0.9,1.1], horizontal and vertical flip, rotations, and grayscale transform. Increasing the range of the scaling is not beneficial in our case since biomarkers of some retinal conditions can be localised anywhere. For instance, microaneurysms can appear anywhere in mild diabetic retinopathy. Additionally, more invasive cropping could result in losing the relevant area. With CLAHE as a preprocessing step, we observed that the training performance decreases when the color jitter or Gaussian blur are considered as random augmentations.

The training on the contrastive loss is run for 500 epochs on 4 GPUs Tesla V100 with a batch size of 512 per device. The initial learning rate is 0.0005 and the temperature is 0.1. The training of the classification layer was conducted for 30 epochs with a batch size of 128.

Applying contrastive learning with the above settings on the Resident dataset leads to the performance metrics in Table 4. Note that the model correctly classifies myopia (C8), vitreous degeneration (C10), branch retinal vein occlusion (C11), rhegmatogenous retinal detachment (C22), cataract (C25), central serous chorioretinopathy (C27), and central retinal vein occlusion (C32), for which the precision is higher than 0.84. We observed that the majority of misclassified examples belong to the mild nonproliferative diabetic retinopathy (part of the DR class), for which the model predicts them to be normal. We observe a higher error rate between AMD (C1) and drusen (C14). One possible explanation is that drusen is the main biomarker in AMD, and hence there is strong association of AMD with different types of drusen, such as hard, soft, small, or large drusen. The worst performance is achieved for laser spots (C29) and tessellated fundus (C9), with one plausible explanation being the fact that they tend to appear together with other pathological conditions.

### 2.3. Assessing Model Performance

There are several possible causes of variable performance of the classification model.

One source of the various levels of diagnostic accuracy is the fact that three datasets are integrated into the Resident dataset. Even though the mapping between classes is conducted by ophthalmologists, there is no perfect alignment between the datasets. For example, the large optic cup (C13) is not among the keywords from ODIR; this does not necessarily mean that there are not any CFPs from the ODIR dataset which have a large optic cup, but all CFPs from ODIR will be considered without C13 in the Resident dataset. A similar situation is for C9 (tessellated fundus), which appears only once in ODIR, while in the RFMID dataset it appears 251 times. C14 (drusen) is missing from JSIEC but it is present in RFMID and ODIR. Glaucoma has positive examples only from ODIR, since in the other two it was considered glaucoma together with suspected glaucoma. Furthermore, in the ODIR and JSIEC datasets, the *normal* class is present, while it is missing from RFMID dataset. The fact that none of the considered conditions are present does not mean that the CFP is normal, since other unconsidered conditions could be present.

Another source is the diversity of the classes: some of them are pathological conditions (e.g., AMD, glaucoma), while others are only changes of the normal aspect without being pathological conditions (e.g., tessellated fundus, laser spots). These tend to appear together with other conditions: for example, in the subset from the ODIR for the training set, out of 33 CFPs with laser spots, 27 are also with moderate non proliferative retinopathy (Table A1).

A third source is the overlapping between two pathological classes, as it happens for C1 (AMD/maculopathy) and C14 (drusen). This adds to the already mentioned problem of the C14 (drusen) class being absent from JSIEC. Consequently, the majority of the errors on these two classes are: (i) either CFPs annotated as only C1 are identified by the model as only C14, or (ii) CFPs annotated as C14 are considered normal or without any condition.

Lastly, there are conditions easy to be identified from the image (e.g., pathological myopia), but also conditions that strongly depend on structure (e.g., glaucoma identification relies on optic disc structure). In literature, glaucoma is usually detected with deep learning only after first applying optic disc [5] or blood vessel segmentation. Furthermore, the difference between normal and mild diabetic retinopathy can stand in only one microaneurysm. For this task, Cen et al. have used images with a double size [5]. Even so, the performance of mild diabetic retinopathy is smaller compared to the other stages of diabetic retinopathy.

To overcome possible classification errors, for the first phase of the implementation, we plan to have an expert physician in the loop, who will validate the DL model’s diagnosis before it is passed to the expert-system-powered case allocation algorithm. In this way, we can evaluate the DL model’s performance and assure a proper assignment of the cases. As real patients come to the retina clinic, their CFPs will be recorded and the DL model will be retrained once more data is available.

Even in the presence of the classification error, residents can benefit more from the case allocation compared to a random allocation of cases, since the largest errors tend to be present in classes that do not represent pathological conditions. Moreover, when the ophthalmologist assesses the answer of the resident, they also diagnose the case. When this correct diagnostic is updated in the system, the case allocation algorithm will take into consideration that the resident has not actually received a case of that type. Hence, the chances of residents receiving all cases are not affected by such misclassifications of the deep learning model. Furthermore, the evaluation of the resident’s performance is based on the diagnosis made by the expert physician, not the diagnosis given by the model.

### 2.4. Automatically Assessing Difficult Cases

To better match residents with cases, one needs to have an a priori estimation of how difficult it would be for the resident to diagnose an image. To assess the case difficulty, we introduce a metric based on two components. The first component (*prob_difficulty*) considers how confident the model is about the predicted classes and to what extent the signs of other classes are identified without enough confidence to predict that class. Let $p_{C}\left(x\right)$ be the model prediction, for example *x*, to belong to the class *C*. If $p_{C}\left(x\right)\geq 0.5$ then *x* belongs to class *C*. An image *x* can belong to more than one class (i.e., multi-label classification). In case more than one class is predicted, the difficulty starts from 1. Otherwise, it starts from 0. For each predicted class, the difference between 1 and the predicted probability $p_{C}$ is added to the difficulty. For each other class predicted with $0\leq p_{C}\leq 0.5$, the probability $p_{C}$ is added to the difficulty.
(1)$$prob\_difficulty\left(x\right)=several+\sum_{p_{C}\left(x\right)\geq 0.5}\left(1-p_{C}\left(x\right)\right)+\sum_{p_{C}\left(x\right)\leq 0.5}p_{C}\left(x\right)$$

The second component (neigbors_difficulty) is based on neighbor images, and it assesses the variety of the images most similar to the classified one (i.e., do they all belong to the same class or to other classes?). The similarity between images is computed as the cosine similarity between embeddings built by the encoder network. This neighbor-based component is inspired by the Silhouette metric, usually used for assessing the quality of clusters. For one image *x*, we identify the most similar *k* images from the Resident test set. In our experiments, we consider k=9. Let us consider that the predicted class for image *x* is *C*. If all the neighbors belong to *C*, then we consider *x* an easy example. If none of the neighbors belong to *C* and they are very close to *x*, the image *x* has a maximum difficulty. If there are examples among neighbors which belong to *C*, but also examples from other classes, the difficulty depends on how well these are examples split into clusters: if the mean distance from *x* to neighbors from the same class is smaller than the distance to neighbors from other classes, it means that even though there is a diversity among the neighbors, the dominant ones are from *C* and the example is rather easy. In contrast, if the mean distance from *x* to neighbors from the same class is larger than the distance to neighbors from other classes, the example is rather difficult.

To formalize this, let $N^{C}$ be the set of neighbor examples that have a common class with the predicted ones, while $N^{C_{j}}$ are neighbors which belong to class $C_{j}$ but do not belong to the predicted classes. The value *a* is the mean of distances from *x* to neighbors which have common classes with *x*:(2)$$a=\frac{1}{|N^{C}|}\sum_{e\in N^{C}}d(x,e)$$

The value *b* is the minimum mean of distances from *x* to neighbors with different classes:(3)$$b=min_{C_{j}\notin C}\frac{1}{|N^{C_{j}}|}\sum_{e\in N^{C_{j}}}d(x,e)$$

Hence, the neighbor-based component is given by the following equation, with $C_{min}$ as the class at distance *b* to *x*:(4)$$Neighbors\_difficulty\left(x\right)=-\frac{|N^{C}\left|*b-\right|N^{C_{min}}|*a}{|N^{C_{min}}|*max\left(b,a\right)}$$

We changed the Silhouette metric to include the dimension of the “positive” cluster and the “negative” closest cluster, where the “positive” cluster includes the neighbors with the same classes as the predicted ones. A good “split” means a large distance between the positive and the negative cluster, but also more positive examples than negative examples. A “good” split of neighbors means a smaller difficulty, hence the minus sign from Equation (Equation 4).

In order to compute the difficulty of one example, both difficulties are normalised to the interval [0,1]. Their sum gives the difficulty of one example. If Difficulty(x)≤0.5 then the example is *easy*, if 0.5≤Difficulty(x)≤1.5, then the difficulty is *medium*, and finally, if 1.5≤Difficulty(x) then the example is *difficult*.
Difficulty(x)=normalized(prob_difficulty)+normalized(neighbors_difficulty)

Having (i) a model trained on the pedagogical Resident dataset to classify new cases in 19 retinal conditions, and (ii) a metric to assess the case difficulty, we next describe the algorithm that allocates cases to residents.

## 3. Case Allocation Algorithm

### 3.1. Problem Statement

The task to solve is: *“Given a case (i.e., patient with a retinal condition), which resident would benefit the most from it?”*. For matching residents with the available cases during the retinal study module, we devised a case allocation algorithm. The algorithm is based on the following assumptions:

First, based on the previous years’ history we assume an expected number of 1.000 patients with retinal conditions visiting the Ophthalmology Department of the Emergency County Hospital, Cluj-Napoca, Romania during one year. These patients need to be assigned fairly to a number of 10 residents within the retinal study module.

Second, we assume 19 main retinal conditions encountered by residents during the training, ordered by their support in the Resident dataset: normal (C0), age-related macular degeneration (C1), diabetic retinopathy (DR—from C2, C3, C4, and C5), glaucoma (C6), hypertensive retinopathy (C7), pathological myopia (C8), tessellated fundus (C9), vitreous degeneration (C10), branch retinal vein occlusion (C11), large optic cup (C13), drusen (C14), epiretinal membrane (C15), optic disc edema (C18), myelinated nerve fibers (C19), rhegmatogenous retinal detachment (C22), refractive media opacity (C25), central serous chorioretinopathy (C27), laser spots (C29), and central retinal vein occlusion (C32) (recall Table 2).

Third, based on the pathologically involved structures and the localisation on the CFP, we group these retinal conditions in six educational topics (see Table 5). We expect that during one year each resident encounters at least three cases from each of the 19 retinal conditions in the Resident dataset. To distribute these three cases during the entire year, we rely on the behaviorist and cognitive learning theories. Note that if residents have difficulties in handling a case, they should receive more cases of the same retinal condition or educational topic.

Fourth, residents should receive a fair number of easy, medium, or difficult cases. We rely on deep learning machinery and on the difficulty function to pre-assess each case according to its difficulty level.

Fifth, the cases are allocated based on the resident’s case history and previous performance. For each case, residents are assigned to examine patients, to make the diagnosis and differential diagnosis (i.e., to explain and support their decisions based on the relevant fundus features), and to make the therapeutic plan.

### 3.2. The Assignment Flow

The retina module lasts 12 months. In the first 2 months (the initiation period), residents become familiar with the CFPs from the Resident dataset and the Objective Structured Clinical Examination (OSCE) checklists for all of the 19 RCs. During this initiation period, the residents are not evaluated. In the following 10 months, every resident will be assigned to examine at least one patient every day (at least 240 patients/year). The performance of residents is evaluated by the expert physician using an OSCE checklist. The checklist evaluates four parameters: (1) current diagnosis $D_{r}$, (2) list of signs (identified and missed) $S_{r}$, (3) list of differential diagnoses (identified and missed) ${\mathrm{DIF}}_{r}$, and (4) patient management $M_{r}$. If residents state the correct diagnosis, they receive one point for each box checked. If residents fail to indicate the correct diagnosis, they receive 0 points. The expert physician then calculates the percentage of correct answers and scores the resident’s performance: 0–20% = 1; 21–40% = 2; 41–60% = 3; 61–80% = 4; 81–100% = 5. At the end of the encounter, the expert physician offers feedback and learning suggestions. The OSCE for diabetic retinopathy is exemplified in Table 6. Note that there is such a checklist for each retinal condition in the Resident dataset. The residents’ grades are recorded in their teaching files and used for assigning future cases. The case difficulty will be decided by the DL model and will be scored as follows: 1 = easy; 3 = medium; 5 = difficult.

The educational goal is that by the end of the retina study module, each resident obtains at least a score of 4 or 5 for each of the 19 RCs on a case with a difficulty of 3. That would equal to a final grade of ≥7, which is calculated based on the resident’s performance score (1 to 5) added to the case difficulty score (1, 3, or 5).

We employ the following case assignment flow (Figure 2):Patients arriving at the retina clinic are examined by a technician, who performs a CFP.The resulting image is analysed by the deep learning algorithm, which generates a presumptive diagnosis and a difficulty score.The diagnosis is validated by the expert physician.The case is sent to the expert system-powered allocation algorithm which will assign the case to a resident.In case of patient shortage addressing to the retina clinic, the allocation algorithm will select a CFP from the test set from the pedagogical Resident dataset, so that every resident is assigned at least one patient or case per day.The resident examines the patient, and based on the clinical signs, elaborates the diagnosis, differential diagnosis, and therapeutic plan.The resident’s performance is evaluated and scored by the expert physician.Residents’ performances are recorded in their teaching file for the specific retinal condition.The pedagogical Resident dataset is extended with the case.

### 3.3. Allocation Rules

For choosing whether to assign real cases (patients from the retina clinic) or virtual cases (CFPs from the pedagogical Resident dataset), the allocation algorithm follows the flow in Figure 3. The top level requirement is to assign to each resident at least one case per day. Algorithm A1 prioritises the assignment of real cases, and in case of a shortage, virtual cases from the Resident dataset. If there are more real cases than residents, some residents may receive more than one case. For real cases, we apply the subsequent rules from Table 7 and Table 8 displayed in Figure A1, while for virtual cases, the following rules in Table 9 are visible in Figure A2. If the educational goals are not met at certain time points, supplementary virtual cases are assigned based on the rules from Table 10 and shown in Figure A3. The classes with retinal conditions removed for insufficient cases will be labelled as “supplementary rare”. Residents who meet the three educational goals set within the paper will receive one virtual supplementary case from these “supplementary rare” classes daily until they have seen one case from each class.

The resident’s grade for a correct diagnosis is composed of the performance score evaluated by the expert physician (1 to 5) plus the difficulty score evaluated by the DL model (1, 3, or 5). The resident’s grade for an incorrect diagnosis is 0.

The resident teaching file records the following information for each of the 19 RCs: (1) number of cases examined, (2) grades for every case encountered and average grade for a retinal condition, (3) time since last encounter of the retinal condition. After every patient, the resident teaching file is immediately updated. Each resident has access to the personal information in the teaching file.

The patients with a RC not contained within the 19 RCs will be assigned following the rules in Section A.1, Figure A1, right side of the figure. First, the algorithm takes into consideration the resident with the fewest seen cases of any RC that day. If the rule applies to more than one resident, the second rule is followed, which considers the resident with the fewest seen cases overall from all RCs. If the rule applies to more than one resident, it is randomly assigned between them. The case will also be assigned as a virtual case to all the other residents in the program, since we consider it a rarer case. The case will not be evaluated and it will not have a difficulty score, neither for real nor for virtual assignments.

If a classification error occurs and for example, a resident receives a case the DL model pre-assessed as hypertensive retinopathy, which in reality is diabetic retinopathy, we have the following scenario: after the evaluation by the expert physician which states that it was correctly identified as diabetic retinopathy, the case is filed in the resident’s teaching file and recorded as a diabetic retinopathy case (updating the number of cases seen, the last case seen, and average performance). Therefore, the expert-system-powered case allocation algorithm looks at the teaching file when assigning cases and will also record it as a diabetic retinopathy case.

### 3.4. Running Scenario

The rules listed above will be used by the allocation algorithm’s expert system. We exemplify here a running scenario of the expert system matching algorithm. The rules were implemented in CLIPS (C Language Integrated Production System [8]). CLIPS is a multi-platform rule-based programming language developed by NASA and made available as public domain software. This choice was made based on the level of complexity of the rules involved in the proposed algorithm, as well as on the need to accommodate future developments. We assume a number of patients present themselves at the hospital during a specific day, one at a time. No knowledge about the order of their arrival is available. A number of residents should take care of them. Each patient will be referred to one of the residents. In this process, we will take into consideration the presumptive diagnosis obtained via the DL component and also the residents’ grade for each condition.

The rule-based system is run for each patient and results in assigning one resident to each patient, according to the algorithm above. We assume three residents, eight patients, and one condition. We start by defining the templates to store data about the available cases and residents (Listing A1).

We also need to store data about the assignments made so far (Listing A2).

Then, we formalise the allocation rules, with an example in Listing A3. The priorities among rules are handled by the salience construct of the CLIPS language. For all current cases (?fcc variable in line 102) and residents (?fm variable in line 104), the system checks whether the number of assignments ?na is less than the maximum value for the current instance. If there is no resident with fewer cases of the given retinal condition, an assignment is asserted in the knowledge base (line 110). The assignment matches the four variables: the case, data, the resident, and the patient. The current case is retracted from the set of non-allocated cases (line 111), while the system keeps track of the number of cases of a specific retinal condition assigned to the resident.

Let the case in which the learned model has computed a presumptive condition *c* (line 200 in Listing A4). Assume the three residents from lines 201–206. Note that resident m503 already has three assignments but only one for the current condition *c*. Resident assignments for each patient appear in lines 208-205. The case 1001 is allocated to resident m503.

## 4. Discussion and Related Work

### 4.1. Case Allocation

At the moment, we designed and implemented a prototype of the system. The ongoing action aims to operationalize the system through a pilot program at the Ophthalmology Department of the Emergency County Hospital in Cluj-Napoca. We consider implementing the following protocol: to enroll 10 to 15 residents in the program and follow them over a period of one year. Every 3 months, evaluate them and listen to their feedback regarding the program through a questionnaire. We should keep in mind that they are also evaluated in the program, after each case with a RC within the 19 RCs. At the end of the one-year program, we compare their performance with that of non-participating residents, through a theoretical and practical evaluation. We will encourage residents not participating in the program to keep records of the cases encountered in the clinic and compare the distribution of their cases with that of the residents involved in the program.

Real patients are an invaluable resource for learning; thus, we decided to fairly assign them to the residents. Virtual cases fill the gap left open by the shortage of real patients. This might also help dealing with rarer RCs not encountered during the rotation.

Our case allocation algorithm integrates the spacing principle. The “spacing effect” phenomenon, identified by the German psychologist Hermann Ebbinghaus in 1885, shows that learning is more effective when practice sessions are spaced out [9]. Spaced repetition algorithms are used to make practices more effective by learning applications such as Duolingo (language learning) or Anki (flash card learning). Duolingo and Anki use the spacing effect by increasing or decreasing the interval between practices based on the user’s performance in that particular area of study. In other words, the learning application will present information to a user more or less frequently if they do or do not show difficulty in remembering it, respectively. In order for the information to be retained longer, there should be a proportional increase in the time between the spaced intervals [10]. Spaced repetition is indeed more beneficial than the massed practice, as shown by this meta-analysis, which takes in consideration 839 assessments [11]. When looking at intersession lags, increasing the space between study sessions helps in long-term retention.

When asked about their study methods, the majority of medical students who took the United States Medical Licensing Examination (USMLE) Step 1 exam reported using self-initiated retrieval practices often accompanied by spaced repetition. Those who reviewed more cards in the flashcard learning program Anki; performed better on the examination [12]. Although the outcomes of spaced repetition were studied mostly on undergraduates, there is evidence of its value for trainees, as well. Spaced repetition testing on osteoporosis care and fracture prevention showed not only better results in residents’ long-term information retention but also an improvement in their patients’ outcomes [13]. In the case of resident training, the spaced repetition algorithm could be used by assigning a particular case to a trainee who has had trouble in providing the correct diagnosis for similar cases in the past. This way, each trainee will be chosen to work more on the areas in which they do not excel.

The allocation algorithm proposed here gradually assigns to every resident 1, 2, and 3 cases of every retinal condition. Afterwards, the residents are assigned cases with retinal conditions where they failed. After meeting the educational goal (i.e., grade ≥ 7), the cases are assigned by taking into account the time of encounter of different retinal conditions.

Kornell and Bjork have looked at the outcomes of massed versus interleaved category learning. Even though participants preferred learning one category at a time, they retained more by learning examples from multiple categories in the same session. For our current research, this would be reproduced as having the resident/trainee encounter different retinal conditions during a working day to improve their long-term information retention [14].

Larsen et al. have compared the effects of spaced repetition studying vs. spaced repetition testing with feedback from a randomised clinical trial, looking at residents’ retention of information 6 months after a didactic conference [15]. It seems spaced repetition testing, when accompanied by feedback, yields better results in long-term retention.

In our retina study module, we apply both principles, spaced repetition and spaced testing. Residents are tested and receive feedback after each case. Even though spaced learning and spaced testing are two valuable tools which are recognised to provide a more robust long-term memory, when learners were evaluated, they were often not aware of which strategies could benefit their memory in the long run [16].

The optimal lag-time between training sessions and testing sessions remains an open question. Similar to the approach of Morin et al. [17], our system has access to the OSCE grading files, and therefore one can monitor the performance of each individual. With such data available on performance tracking, a learning system can determine the efficacy of the allocation task both in terms of the allocated cases and allocation time. Hence, one can determine the ideal number of allocation cases and the lag-time between review sessions.

Increasing the number of cases that radiology residents are exposed to correlates with a boost in their performance. However, once a certain volume is passed, Agarwal et al. have observed that the performance starts to decline [18]. This has also been demonstrated by Liebman et al. with surgical exposure in cataract surgery [19]. AI methods could help in finding the optimal exposure balance. Our case allocation algorithm first aims to meet the educational goal and then to fairly allocate cases of different variety and difficulty to boost the resident performance on all retinal conditions.

The current matchmaking task is somewhat different from the classical fair matchmaking tasks, which assume that items are given in advance. In our retina clinic, the patients arrive at the clinic one by one with different retinal conditions. This allocation problem is similar to selecting donor organs to patients, donating food to charities, connecting electric vehicles to charging stations, and distributing water rights to farmers [20].

A similar approach for the personalised case distribution for residents was suggested in the sphere of radiology. Duong et al. [21] proposed as a future solution an AI-based model which could supervise the distribution of imaging cases to residents based on their learning profiles, help them with drafting reports, and file the imaging studies inside their teaching file.

### 4.2. Resident’s Evaluation and Teaching File

During residency training, numerous post-graduate programs require case logs to evaluate the resident’s experience. We view our retinal teaching file as an enhancement to resident case logs, where, in addition to assessing the experience and performance, they also provide grounds for personalised case assignment.

The OSCE examination is a valuable tool for the standardised evaluation of clinical performance. It is largely used in medical education for both undergraduate and post-graduate evaluation [22]. The residents’ performance after each case is evaluated and scored from 1 to 5 by an expert physician using an OSCE file. The final grade is calculated by adding the case difficulty score (1 = easy, 3 = medium, or 5 = difficult) appreciated by the DL model. The educational goal is that by the end of the retina study module, each resident receives a grade ≥ 7 for all the 19 RCs, i.e., a score of 4 or 5 on a case of medium or high difficulty. Grading residents using the OSCE files is based on both the behaviorist learning theory and the cognitive learning theory. The OSCE score is 0 when the correct diagnosis is missed. If the diagnosis is correct, the score is calculated in light of the cognitive theory based on the percentage of correctly identified clinical signs, differential diagnosis, and therapeutic management in the OSCE file (0–20% = 1; 21–40% = 2; 41–60% = 3; 61–80% = 4; 81–100% = 5).

There is numerous evidence [23,24,25] which shows the power of self-testing or retrieval-based learning during individual study. This produces the most effective long-term retention of the material. For instance, during a retrieval-based learning with flashcards, answering correctly more than once, also known as high criterion learning, shows stronger short- and long-term recall capacities [24]. In the same vein, our database is accessible for self-training and self-testing until the end of the residency program. The CFPs of the patients visiting the retina clinic for diagnosis and treatment are analysed by the DL model. The model outputs a presumptive diagnosis and a case difficulty score, which is then validated by the expert clinician. Finally, the patients’ CFPs along with metadata are included in the Resident  dataset.

### 4.3. AI for Education

The learning model’s diagnostic capacity is limited to the data found in the publicly available datasets. We therefore selected the most commonly encountered RCs, with the highest educational impact and supported by sufficient data. Future work involves expanding the model’s training set, both in terms of class variety and size, by using public and also in-house datasets. Unfortunately, there still are not sufficient public datasets globally for public access. Khan et al. [26] have made a substantial effort in collecting and reviewing those available to the public. Therefore, the European Health Data Space is expected to boost AI applications in the medical domain, including those for residency training and precision education.

In order for trainees to fully grasp the potential of AI-guided medical education, future medical school curricula and residency programs will need to integrate an introductory basic understanding of AI [27].

The base use case of our AI system for education is to use a DL algorithm (i.e., contrastive learning) to provide a presumptive diagnosis and difficulty score, based on which the case is assigned to a resident. However, the current AI technology can go beyond this base case, by addressing more fancy cases:(i)Can AI (e.g., machine learning or Bayesian networks) identify the most similar cases?(ii)Can data augmentation generate more cases to train the resident?(iii)Can AI identify clinical features?(iv)Can AI provide a differential diagnosis (e.g., ranking diagnosis based on their probability?)(v)Can AI assess how incomplete input (e.g., missing features) may lead to erroneous interpretations?

### 4.4. Towards Compliance with the Artificial Intelligence Act

The Artificial Intelligence Act (AI Act) is a pioneering regulatory approach from the European Commission to ensure that AI systems are developed in-line with European values. The AI Act has a risk-based approach, in which the AI systems are classified within four classes: (i) prohibited practices, (ii) high-risk areas, (iii) low risk, and (iv) minimal risk. Four practices will be prohibited within the European Union: (i) Remote biometric identification in publicly accessible spaces; (ii) Social scoring of natural persons; (iii) AI-enabled manipulative techniques; (iv) Exploit vulnerabilities of children and people from a specific group of persons. High-risk AI systems listed in Annex III of the AI Act include eight areas: (i) Biometrics; (ii) Critical infrastructure; (iii) Education and vocational training; (iv) Employment or workers management; (v) Access to essential private and public services; (vi) Law enforcement; (vii) Migration, asylum, and border control; (viii) Administration of justice. Note that the third area refers to AI systems applied to education training, and our system for improving residents’ training is in this area.

At the time of our system’s development, the text from the AI Act is updated and negotiated with the member states and stakeholders. The current version (3 November 2022), representing the fifth compromise proposal of the AI Act, has the following text regarding AI systems for education and vocational training: (1) “AI systems intended to be used to determine access, admission or to assign natural persons to educational and vocational training institutions or programmes at all levels”; (2) “AI systems intended to be used to evaluate learning outcomes, including when those outcomes are used to steer the learning process of natural persons in educational and vocational training institutions or programmes at all levels”. First, our system is an AI system since it relies on machine learning and expert systems, and these two technologies are explicitly mentioned by the AI Act as AI technologies. Second, the system does not seem to be referred to by the text from the AI Act, for it does not evaluate the learning outcomes of residents. In our design, the expert ophthalmologist is responsible for the grading of the student, while the AI system is used only to allocate the cases. Nevertheless, we are aware that in order to be used in practice for resident training, the system needs to pass various validation tests for performance, fairness, and trust. With this aim, we continue to follow the guidelines for the high-risk AI systems for education and vocational training.

The high-risk systems will follow a conformity assessment procedure by a third party or certification body. During this procedure, the high-risk systems will require to prove conformance with the AI Act requirements and implementation guidelines regarding risk management, data governance and quality, technical documentation, human oversight, post market monitoring, logs, quality management, transparency, accuracy, or robustness. We aim to address part of these aspects as follows.

*Post market monitoring* is relevant to our system across two dimensions, at least. First, the educational Resident dataset is continuously enriched with new real cases diagnosed by residents and validated by expert ophthalmologists. This enrichment will propagate through the contrastive learning step and case difficulty assessment. Hence, there is a need to check that the system will meet the initial quality specifications. Second, the devices used to take images are continuously improving. Hence, there is the need to assure that the newly added images can be correctly classified by the updated model.

An argument for the *robustness* of the model is that the Resident dataset was obtained by combining three different datasets in which the images were collected from different time periods, with different cameras, from different institutions, and were annotated with different protocols. The test set is also a combination from the three public datasets. However, further experiments are needed to assess the vulnerabilities of the model and to compute robustness evaluation metrics.

*Transparency* is partly achieved by the fact that the allocation algorithm uses rules formalised in the CLIPS expert system. Hence, the reasoning steps can be traced and explained to the human agent.

Since the *technical documentation* should facilitate the verification of algorithm performance, it can be based on assurance argumentative patterns. These assurance patterns are adopted for the medical domain in line [28] and they can be modelled with the Goal Structuring Notation (GSN) [29]. The GSN language provides six types of nodes: (i) Goals to state what to assure; (ii) Contexts to specify state, environment or conditions of the system; (iii) Strategies to describe how to break down a goal into subgoals; (iv) Evidence to assure the goal can be reached; (v) Monitoring to represent evidence available at runtime; (vi) Undeveloped nodes to indicate the status of no evidence or monitoring supporting the goal. Safety cases built with the GSP standard are helpful to structure evidence and therefore to facilitate the technical audit of AI systems in the medical domain.

## 5. Conclusions

AI has been applied for optimizing the clinical training of ophthalmology residents during the retina study module. During this module, we proposed an expert-system-powered allocation algorithm for matching cases with residents. The algorithm interleaves pedagogical instruments (e.g., OSCE checklists) with clinical constraints, aiming to achieve precision education in healthcare. Technically, the allocation algorithm relies on deep learning and expert systems. The learning component uses contrastive learning to compute the presumptive diagnosis and its difficulty. The expert component is used to formalise rules for matching residents with cases.

We advanced the state of the art in the following directions:

First, we built the pedagogical Resident dataset for retinal conditions. The Resident dataset was built by combining three public datasets (ODIR, RFMID, and JSIEC) containing CFPs. To handle the heterogeneity among the three datasets, mapping rules were devised by expert ophthalmologists. To meet the quality criteria required by machine learning algorithms, the dataset has filtered such data to contain a relevant number of training and testing instances for each class. The resulting Resident dataset contains 9.693 fundus images, classified in 19 retinal conditions.

Second, we apply contrastive learning to compute a presumptive diagnosis. Learning from and validating 7.754 images from the Resident dataset, the performance on the testing set of 1939 instances indicated that out of 19 conditions, 11 conditions can be predicted with precision above 0.7, and 10 conditions with a recall above 0.7 in a multilabel setting.

Third, we defined a function to automatically assess the difficulty of a case. The function considers both global and local aspects since it combines the model’s confidence in its prediction with a measure of the diversity of examples that are in close proximity to the classified example.

Fourth, we designed a fair allocation algorithm to match the residents with the cases. The expert system algorithm takes into account the resident’s case history and grades, and its main priority is to help each resident achieve their educational goals. It assures equal exposure to cases and leads residents to improve their performance in diagnosing and managing patients with RCs. Technically, the algorithm has implemented production rules from expert systems.

Interested researchers can extend or improve the current version at least on the following lines: (1) One line would be to improve the DL model to output the list of signs present in an image, facilitating both the resident (who could use it as an aid) and the physician (who could save time during the OSCE evaluation); (2) Another line would be to consider an algorithm that automatically grades the resident, instead of the physician, as in our approach. The algorithm could be designed to include elements from the constructivist learning theory. For instance, it could consider the time it takes residents to complete the OSCE files. In this scenario, we still think it would be valuable to have feedback from the expert physician.

The proposed solution avoids some drawbacks in the current trainee education in healthcare. The residents benefit by receiving: (i) A daily case (real or from the Resident dataset) to solve; (ii) Personalised cases based on their needs with better coverage of the disease spectrum; (iii) Cases with a level of difficulty adapted to their current performance; (iv) A more uniform distribution of similar cases during the one year of practice; (v) Feedback from the expert physician.

In order to have global acceptance and implementation of AI precision education in ophthalmology trainee education, further work is needed to evaluate the trainees’ experience to prove its efficacy. In the proposed educational model based on AI, two processes intertwine: training and testing of residents and training and testing of the DL model.

## Figures and Tables

**Figure 1 jcm-12-01825-f001:**
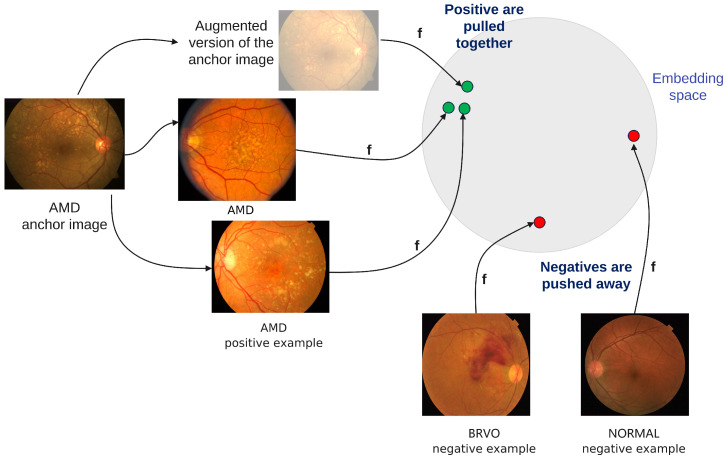
Supervised contrastive learning.

**Figure 2 jcm-12-01825-f002:**
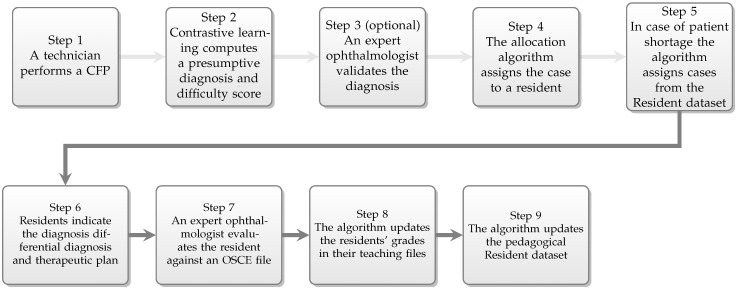
The assignment workflow.

**Figure 3 jcm-12-01825-f003:**
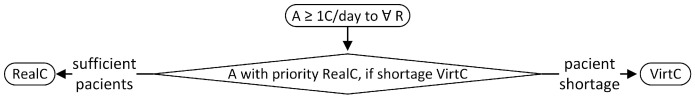
Rules for assigning real or virtual cases (A = assign; C = case; R = resident; RealC = real cases; VirtC = virtual cases).

**Table 1 jcm-12-01825-t001:** Combining public datasets.

Dataset	CFPs	Device	Conditions
ODIR	10,000	Canon, Zeiss, Kowa	8
RFMID	3200	TOPCON 3D OCT-2000, TOPCON TRC-NW300 and Kowa VX-10α	45
JSIEC	1000	ZEISS FF450 Plus IR Fundus, Topcon TRC-50DX Mydriatic Retinal Camera	39
Resident	9693		19

**Table 2 jcm-12-01825-t002:** Mapping rules for combining ODIR, RFMID, and JSIEC into 39 classes (C0–C38). Classes indicated with Gray have been removed since they contain too few instances (<50).

New Class	ODIR Keywords	RFMID Class	JSIEC Class
C0	Normal fundus		0.0.Normal
C1	Age-related macular degeneration Dry age-related macular degeneration Wet age-related macular degeneration	ARMD	6.Maculopathy
C2	Mild nonproliferative retinopathy		0.3.DR1
C3	Moderate non proliferative retinopathy		1.0.DR2
C4	Severe nonproliferative retinopathy		
C5	Proliferative diabetic retinopathy severe proliferative diabetic retinopathy		1.1.DR3
C6	Glaucoma		
C7	Hypertensive retinopathy		
C8	Pathological myopia	MYA	9.Pathological myopia
C9	Tessellated fundus	TSLN	0.1.Tessellated fundus
C10	Vitreous degeneration	AH	18.Vitreous particles
C11	Branch retinal vein occlusion	BRVO	2.0.BRVO
C12			5.1.VKH disease
C13		ODC	0.2.Large optic cup
C14	Drusen	DN	
C15	Epiretinal membrane Epiretinal membrane over the macula Macular epiretinal membrane	ERM	7.ERM
C16		TD	13.Dragged Disc
C17			14.Congenital disc abnormality
C18	Optic disc edema	ODE	12.Disc swelling and elevation
C19	Myelinated nerve fibers	MNF	17.Myelinated nerve fiber
C20			15.1.Bietti crystalline dystrophy
C21			16.Peripheral retinal degeneration and break
C22	Rhegmatogenous retinal detachment		4.Rhegmatogenous RD
C23	Macular hole	MHL	8.MH
C24	Chorioretinal atrophy	CB	24.Chorioretinal atrophy-coloboma
C25	Cataract Refractive media opacity	MH	29.0.Blur fundus without PDR
C26			29.1.Blur fundus with suspected PDR
C27	Central serous chorioretinopathy	CSR	5.0.CSCR
C28			19.Fundus neoplasm
C29	Post laser photocoagulation, laser spot	LS	27.Laser Spots
C30			28.Silicon oil in eye
C31	Central retinal artery occlusion Branch retinal artery occlusion (BRAO)	CRAO BRAO	3.RAO
C32	Central retinal vein occlusion	CRVO	2.1.CRVO
C33	Retinitis pigmentosa	RP	15.0.Retinitis pigmentosa
C34		PRH	25.Preretinal hemorrhage
C35		EDN	20.Massive hard exudates
C36			21.Yellow-white spots-flecks
C37		CWS	22.Cotton-wool spots
C38		TV	23.Vessel tortuosity

**Table 3 jcm-12-01825-t003:** Resident dataset: Numerical description of the train and test sets.

	Train Set					Test Set			
**#**	**Label**	**ODIR**	**RFMID**	**JSIEC**	**Total**	**ODIR**	**RFMID**	**JSIEC**	**Total**
1	C0	2270	0	28	2298	601	0	10	611
2	C1	88	138	0	226	23	31	0	54
3	C2	441	0	0	441	95	0	0	95
4	C3	756	0	38	794	180	0	11	191
5	C4	129	0	0	129	28	0	0	28
6	C5	25	0	29	54	4	0	10	14
7	C6	209	0	0	209	48	0	0	48
8	C7	157	0	0	157	30	0	0	30
9	C8	195	126	40	361	48	35	14	97
10	C9	1	204	9	214	0	47	4	51
11	C10	53	18	11	82	14	1	3	18
12	C11	21	89	29	139	4	27	15	46
13	C13	0	304	36	340	0	69	14	83
14	C14	248	174	0	422	51	53	0	104
15	C15	294	19	22	335	74	5	4	83
16	C18	1	76	12	89	1	20	1	22
17	C19	77	3	10	90	14	0	1	15
18	C22	0	0	44	44	0	0	13	13
19	C25	288	382	84	754	57	104	27	188
20	C27	0	42	9	51	0	19	5	24
21	C29	33	17	16	66	8	4	4	16
22	C32	1	34	15	50	0	10	7	17

**Table 4 jcm-12-01825-t004:** Applying contrastive learning on the Resident dataset. The diabetic retinopathy class (DR) collects all 4 stages of the disease from classes C2, C3, C4, and C5.

Label	Precision	Recall	F1-Score	Support
C0	0.75	0.79	0.77	611
C1	0.67	0.74	0.70	54
C6	0.41	0.29	0.34	48
C7	0.38	0.10	0.16	30
C8	0.87	0.85	0.86	97
C9	0.51	0.55	0.53	51
C10	0.86	0.67	0.75	18
C11	0.90	0.78	0.84	46
C13	0.69	0.86	0.76	83
C14	0.60	0.50	0.54	104
C15	0.63	0.40	0.49	83
C18	0.75	0.68	0.71	22
C19	0.75	0.60	0.67	15
C22	1.00	0.92	0.96	13
C25	0.84	0.84	0.84	188
C27	0.86	0.79	0.83	24
C29	0.43	0.75	0.55	16
C32	1.00	1.00	1.00	17
DR	0.73	0.67	0.70	328

**Table 5 jcm-12-01825-t005:** Educational topics in the retina study module.

#	Educational Topic	Retinal Condition
$T_{1}$	Normal	Normal (C0), tessellated fundus (C9)
$T_{2}$	Macular conditions	Age-related macular degeneration (C1), pathological myopia (C8), drusen (C14), epiretinal membrane (C15), central serous chorioretinopathy (C27)
$T_{3}$	Vascular conditions	Diabetic retinopathy (DR), hypertensive retinopathy (C7), branch retinal vein occlusion (C11), central retinal vein occlusion (C32)
$T_{4}$	Optic nerve conditions	Glaucoma (C6), large optic cup (C13), optic disc edema (C18), myelinated nerve fibers (C19)
$T_{5}$	Peripheral retina conditions	Rhegmatogenous retinal detachment (C22), laser spots (C29)
$T_{6}$	Transparent media conditions	Vitreous degeneration (C10), refractive media opacity (C25)

**Table 6 jcm-12-01825-t006:** Diabetic retinopathy OSCE evaluation file. The performance score corresponds to the following percentages: 1 for 0–20%; 2 for 21–40%; 3 for 41–60%; 4 for 61–80%; 5 for 81–100%.

*Resident:*		*Date:*	
	OSCE DR		
Correct Diagnosis Pass (Calculate Score)	☑	Wrong Diagnosis Fail (0 Points)	☐
*Clinical fundus signs*	*(each box = 1 point)*		
Microaneurysms	☑	Neovascularisation of the disc	☑
Dot-blot hemorrhages	☑	Neovascularisation elsewhere	☐
Hard exudates	☑	Preretinal hemorrhage	☐
Cotton-wool spots	☑	Vitreous hemorrhage	☐
Venous beading	☑	Tractional retinal detachment	☐
Intraretinal microvascular anomalies	☑	Laser spots	☐
*Differential diagnosis of macular edema*	*(each box = 1 point)*		
Hypertensive retinopathy	☑	Macular edema secondary to epiretinal membrane	☐
Central retinal vein occlusion	☐	Ruptured microaneurysm	☐
Branch retinal vein occlusion	☑	Irvine gass syndrome	☑
Choroidal neovascular membrane	☐	Post uveitic macular edema	☑
*Differential diagnosis of retinopathy*	*(each box = 1 point)*		
Central retinal vein occlusion	☑	Valsalva retinopathy	☐
Hemiretinal vein occlusion	☑	Sickle cell retinopathy	☐
Branch retinal vein occlusion	☑	Post-traumatic retinal bleed	☑
Hypertensive retinopathy	☑	Retinal macroaneurysm	☐
Ocular ischemic syndrome	☑	Retinopathy in thalassemia	☐
Terson syndrome	☑		
*Management of macular edema*	*(each box = 1 point)*		
Observation	☐	Intravitreal anti-VEGF	☑
*Management of retinopathy*	*(each box = 1 point)*		
Observation	☐	Intravitreal anti-VEGF	☑
Panfundus laser photocoagulation	☑	Vitrectomy	☐
Resident scored (29) points of a total of 37			
*Physician:*		*Score: (4)*	

**Table 7 jcm-12-01825-t007:** Assignment rules for real cases with a diagnosis contained in the 19 RCs.

#	Rule
$r_{1}$	Assign at least one case/day to each resident
$r_{2}$	Assign with priority patients presenting to the retina clinic, then, in case of shortage, CFPs from the Resident dataset
$r_{3}$	Assign one case from each of the 19 retinal conditions to each resident
$r_{4}$	Assign the case to the resident which has seen fewer cases from this retinal conditions, up to 3 cases
$r_{5}$	Assign the case to the resident with the lowest grade (performance score + difficulty score) until all residents obtain a grade ≥ 7 for every retinal condition
$r_{6}$	Assign the case to the resident with the oldest encounter for that specific condition
$r_{7}$	Assign the case to the resident with the lowest number of cases from that specific educational topic
$r_{8}$	Assign the case to the resident with the lowest number of cases from that specific retinal condition
$r_{9}$	Assign the case to the resident with the lowest number of cases from all the 19 retinal conditions

**Table 8 jcm-12-01825-t008:** Assignment rules for real cases with a diagnosis not contained in the 19 RCs.

#	Rule
$r_{3}^{b}$	Assign the case to the resident who has seen fewer cases overall that day and at the same time, add the case to virtual cases and supplementary assign it as a virtual case to each resident
$r_{4}^{b}$	Assign the case to the resident with the lowest number of cases from all retinal conditions

**Table 9 jcm-12-01825-t009:** Assignment rules for virtual cases.

#	Rule
$r_{3}^{v}$	Assign one case from each of the 19 RCs
$r_{4}^{v}$	Assign the resident a case from the RCs with fewer encountered cases, up to 3 cases
$r_{5}^{v}$	Assign the resident a case from to the RC with the lowest grade (performance score + difficulty score) until a grade ≥ 7 for all retinal conditions
$r_{6}^{v}$	Assign the resident a case from the RC with the oldest encounter
$r_{7}^{v}$	Assign the resident a case from the RC with the fewest cases seen

**Table 10 jcm-12-01825-t010:** Assignment rules for supplementary virtual cases.

#	Rule
$r_{1}^{s}$	If after month 5, there are still residents who have not seen 1 case from each of the 19 conditions, start supplementarily assigning 1 virtual case each day for every resident until the criteria is met
$r_{4}^{s}$	If after month 7, there are still residents who have not seen 3 cases from each of the 19 retinal conditions, start supplementarily assigning 1 virtual case each day for every resident until the criteria is met
$r_{5}^{s}$	If after month 9, there are still residents who have not achieved a grade of 7 or higher on every one of the 19 retinal conditions, start supplementarily assigning 1 virtual case each day for every resident until the criteria is met.

## Data Availability

The Resident dataset was built from three publicly available online datasets: Ocular Disease Intelligent Recognition (ODIR) [2,3]: https://www.kaggle.com/datasets/andrewmvd/ocular-disease-recognition-odir5k. Retinal Fundus Multi-Disease Image Dataset (RFMID) [4]: https://ieee-dataport.org/open-access/retinal-fundus-multi-disease-image-dataset-rfmid. The public part (1000 CFPs) of the Joint Shantou International Eye Centre (JSIEC) dataset [5]: https://www.kaggle.com/datasets/linchundan/fundusimage1000. The CSV file for the Resident dataset can be found at: https://github.com/ancamarginean/personalized_ophthalmology_residency_training, all accessed on 27 December 2022.

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
