# Peer review of "Artificial Intelligence for Personalised Ophthalmology Residency Training"

_jcm, 2023, doi:10.3390/jcm12051825_

Round 1
Reviewer 1 Report
· The use of a metric to assess case difficulty is very interesting and valuable in assigning cases to residents based on experience level. I applaud the authors on this valuable work.
· The ability of the DL model to ‘triage’ the patients based on CFPs can be useful to assign suitable patients to appropriate level of trainees as part of the balance between service and education needs.
· The granularity of CFP diagnosis is important as it will provide different levels of education based on trainee experience. E.g. If diagnosis of ‘wet age-related macular degeneration’ was made, it may encompass varying complexity with barn door CFPs of wet ARMD ideally given to junior trainees and more complex CFPs given to senior trainees for a start.
· Who performed the mapping for building the resident dataset? Ophthalmologists? Retina specialist?
· What happens when the presenting patient has a retinal condition not within the 19 RCs? Does it contribute to added difficulty?
· What was the model performance when compared to diagnosis validated by the expert physician?
· In the scenario where the AI wrongly classifies the patient’s CFP (given a precision of ≥0.7) e.g. AI classified as hypertensive retinopathy, but the student answered diabetic retinopathy which is graded correct by the expert physician – the student will receive a high score. Will the AI then record this as a ‘correctly’ identified CFP in the hypertensive retinopathy RC category, thus the resident will unlikely get a hypertensive retinopathy CFP any time soon? How are such AI model errors managed?
· It is better to demonstrate robustness, rather than to state it is achieved simply by the combination of heterogenous datasets
· The authors should provide more information on the actual performance of the deep learning model.
· Further considerations aside from GAN as rightly pointed out by the authors, may be the consideration of multimodal imaging which is increasingly key in retina diagnostics
Reviewer 2 Report
Authors describe in very thorough way their study- they used AI for optimizing clinical training of ophthalmology residents during the retina rounds.
I think they did wonderful work building the model and applying it into clinical use.
I do think that before publication few changes should be made to the article-
First, there are too many unnecessary details in many places, for example- they can replace the detailed section of 2.1. (Pages 3-5) and present the information only in the tables.
Secondly, there are a lot of unnecessary diagrams, maybe consider transfer them to Supplements? For example, diagrams 4,5 and 6. Diagram 2 is completely unnecessary.
Algorithm 1 and the entire "listing" can also be included only in supplements
If I understand correctly, no results were described...Authors should consider giving some information regarding how many residents concluded the program, what was their grades? Did they fulfill some kind of feedback? Did they compare residents' success in final exams to previous residents they were training?
Reviewer 3 Report
At the end of each case, the attending expert physician assesses the resident’s performance based on standardized examination files and the results are immediately updated in his portfolio. Artificial intelligence did not play a major role in this study.
Reviewer 4 Report
The authors adopted machine learning and an expert system to pursue the balanced teaching of residents. For this, the authors developed a deep learning model to differentiate the color fundus photo (CFP) automatically and to give a difficulty score to CFP. Subsequently, a case allocation algorithm was developed to distribute a case to the resident who would most benefit from the specific case. Then, the attending expert physician assesses the resident’s performance based on OSCE examination, and the algorithm updates the resident’s grade.
This approach is very interesting and meaningful because the balanced education of residents is usually ignored during the residency program.
1. In Table 4, the F1 score - harmonic mean of precision and recall – is somewhat low (<0.5) in some labels (e.g., C6, C7, C15). Is there any chance for these labels with low-performance metrics to have the proper allocation of cases?
Do the various level of diagnostic accuracy on CFP come from the innate characteristics of the dataset or the used DL architecture (contrastive learning)?
2. Please give more details of the train the DL model.
- What was the resolution of the input images?
- If the authors augment the input images, what kind of image augmentation did the authors use?
- How many epochs was the DL network trained?
3. When the authors built the resident dataset, they removed some diseases (Table 1; e.g., VKH, MH, RAO) because the three datasets did not contain enough cases. Meanwhile, these relatively rare diseases are also important, and they should also be considered in the educational program. What would be the way to compensate?
Round 2
Reviewer 2 Report
I thank the authors for all the changes' correction and clarifications they have made.
Reviewer 4 Report
The authors adequately answered the raised questions and revised the manuscript.